

# Taxonomic status of *Coryphophylax maximiliani* Fitzinger in: Steindachner, 1867 with notes on *Coryphophylax subcristatus* (Blyth, "1860" 1861)

Zeeshan A. Mirza[1], Saunak Pal[2], Tejas Thackeray[3], Harshil Patel[3] and Aaron M. Bauer[4]

[1] Department of Integrative Evolutionary Biology, Max Planck Institute for Biology, Tübingen, Baden Württemberg, Germany
[2] Newcastle University, Newcastle, United Kingdom
[3] Thackeray Wildlife Foundation, Mumbai, India
[4] Department of Biology and Center for Biodiversity and Ecosystem Stewardship, Villanova University, Villanova, PA, United States

Corresponding author
Zeeshan A. Mirza, zeeshan.mirza@tuebingen.mpg.de

## ABSTRACT

The insular agamid genus *Coryphophylax* Fitzinger *in* Steindachner, 1867, is endemic to the Andaman and Nicobar Islands in the Bay of Bengal, India. These diurnal lizards are currently represented by two putative species, *Coryphophylax subcristatus* (Blyth, "1860" 1861) and *Coryphophylax brevicauda* Harikrishnan, Vasudevan, Chandramouli, Choudhury, Dutta & Das, 2012. The species *C. subcristatus* is said to be distributed through the Andaman and Nicobar Islands, even across the Ten Degree Channel, which is a recognised biogeographic barrier. A reassessment of the taxonomy of *C. subcristatus* shows the population south of the Ten Degree Channel, for which the nomen *Coryphophylax maximiliani* Fitzinger *in* Steindachner, 1867 is available, to be distinct. The results are based on morphological data from museum material, including type specimens and mitochondrial 16S rRNA sequences. The members of the genus *Coryphophylax* are abundant and widespread across the islands and may serve as an illuminating example for studying the patterns of colonization and diversification across the Andaman and Nicobar Islands.

## INTRODUCTION

The agamid genus *Coryphophylax* is endemic to the Andaman and Nicobar Islands, and is represented by two species, namely *Coryphophylax subcristatus* (Blyth, "1860" 1861) and *Coryphophylax brevicauda* Harikrishnan, Vasudevan, Chandramouli, Choudhury, Dutta & Das, 2012 (*Harikrishnan et al., 2012*; *Uetz & Hošek, 2024*). The populations of each species of the genus are diurnal and abundant and are among the more conspicuous components of the lizard assemblage on these islands (*Das, 1999*; *Rangasamy et al., 2018*). The validity of the genus has been a matter of debate as *C. subcristatus* has also been included in the genus *Gonocephalus* Kaup, 1825 (*Smith, 1935*; *Das, 1999*; *Sharma, 2002*;

*Harikrishnan et al., 2012*). The genus was included in a recent broad phylogeny of Draconinae, represented by a single species, and was recovered as sister to the genus *Bronchocela* Kaup, 1827 (*Pal et al., 2018*).

The Andaman and Nicobar Islands are a group of 572 islands that are peaks of submerged hills of the Arakan Yoma range running from Myanmar to Sumatra. The Ten Degree Channel, a >1,000 m deep gorge spanning about 140 km, divides the islands into two groups, namely, the Andaman Archipelago to the north and the Nicobar Archipelago to the south (*Harikrishnan et al., 2012*). The reptilian assemblage of the two island groups suggests that they are distinct from each other (*Das, 1999*; *Rangasamy et al., 2018*; *Rao, Chandra & Devi, 2017*; *Smith, 1941*; *Vijayakumar & David, 2006*). The biota of the Andaman Islands shows affinity to Burmese taxa; in contrast, the Nicobar Islands is considered to have an affinity to taxa from Sumatra (*Das, 1999*; *Harikrishnan et al., 2012*; *Ganeshaiah et al., 2019*; *Chandramouli et al., 2023*). One notable exception is *C. subcristatus*, which has been reported to be distributed across all islands of the Andaman and Nicobar Islands, except Great Nicobar Island (*Smith, 1935*; *Tikader & Sharma, 1992*; *Das, 1999*; *Harikrishnan et al., 2012*). However, the apparent widespread distribution of the species may be an artefact of the lack of critical taxonomic assessment, and re-examination of type material of current putative synonyms may reveal cryptic species more in line with the well supported distinctiveness of the two archipelagos (*Das, 1999*; *Harikrishnan et al., 2012*).

*Coryphophylax subcristatus* was described by Blyth ("1860" 1861) as *Tiaris subcristata*, which was later transferred to the genus *Coryphophylax* erected by Fitzinger *in Steindachner, 1867*. *Fitzinger (1861)* used the name *Coryphophylax maximiliani* for the population of the Nicobar Islands but without an accompanying description. A detailed description based on a series of specimens (traceable specimens NHMW 20976:1–9) housed at the Natural History Museum of Vienna appeared only in Fitzinger *in Steindachner (1867)*, making the nomen available from that date. *Stoliczka (1873)* described *Tiaris humei* from 'Tillinchang' (=Tillingchang, Nicobar Islands, India) based on one male and one female specimen, the former deposited at the Asiatic Society of Bengal (now the Zoological Survey of India, Kolkata) and the latter specimen possibly at the Natural History Museum, London (*Das, Dattagupta & Gayen, 1998*). *Boulenger (1885, 1890)* listed '*humii*' and '*subcristatus*' as valid species but did not provide an account for '*maximiliani*', nor was it included among the synonyms of the other species. The two nomina, '*maximiliani*' and '*humei*' were synonymised with *C. subcristatus* by *Annandale (1904)*, who examined multiple specimens from across the islands and concluded that the specimens from Nicobar were just exceptionally large individuals of *C. subcristatus*. This taxonomic conclusion was followed by subsequent workers (*Smith, 1935*; *Biswas & Sanyal, 1977*; *Biswas, 1984*; *Tikader & Sharma, 1992*; *Krishnan, 2005*; *Harikrishnan et al., 2012*).

The lack of fresh material and the difficulty of procuring permission to work on these islands have been significant impediments to studies on reptiles. In this regard, we examined the material of the genus *Coryphophylax* across natural history collections in Europe and India, and we present preliminary notes on the taxonomy of its members. Results from the investigation led to the revalidation of a putative synonym of *C.*

*subcristatus* and provided evidence for an additional undescribed congener in the Nicobars.

## MATERIALS AND METHODS

### Morphology

The study was based on 20 museum specimens (14 of the Nicobar population and 6 of *C. subcristatus*, and no live individuals were captured or collected for this study. Meristic data was taken with a Mitutoyo™ dial calliper to the nearest 0.1 mm. Morphological data were recorded with an Olympus stereo binocular microscope SZ40. The following morphological characters were recorded following *Sadasivan et al. (2018)* and *Ambekar, Murthy & Mirza (2020)* with slight modifications. The following measurements were taken: snout-vent length (SVL, from tip of snout to anterior border of cloaca), head length (HL, from snout tip to posterior border of tympanum), head width (HW, distance from left to right outer edge of the head at its widest point), head height (HH, dorsoventral distance from top of head to underside of jaw at transverse plane intersecting angle of jaws), snout-eye length (SE, from snout tip to anterior border of orbit), eye to tympanum (ET, from posterior border of orbit to anterior border of tympanum), jaw length (JL, from rostrum to corner of jaw), interorbital width (IO, transverse distance between anterodorsal corners of left and right orbits), nares to eye (NE, distance from the anterior edge of orbit to posterior edge of naris), snout width/internasal distance (IN, transverse distance between left and right nares), tympanum diameter (TD, greatest diameter of tympanum), orbit diameter (OD, distance between anterior and posterior margins of orbit), lower arm length (LAL, distance from elbow to distal end of wrist, or just underside of forefoot when the limb is flexed), upper arm length (UAL, distance from anterior insertion of forelimb to elbow when the limb is flexed), crus length (CL, length of crus (tibia) from knee to heel), hind foot length (HFL, distance from proximal end (heel) of hind foot to distal most point of fourth toe), trunk length (TrL, from forelimb insertion to hind limb insertion), trunk height (TrH, depth midway between the fore and hind limb insertions), trunk width (TrW, width midway between the fore and hind limb insertions), tail length (TaL, from posterior border of cloacal opening to tip of tail), tail height (TaH) and tail width (TaW, at tail base). Meristic characters were counted for multiple individuals per species. The following characters were scored: mid-body scale rows (MBS, number of scale rows around the trunk at midbody), mid-dorsal scales (MD, counted from the first erect dorsal crest spine to the level above the vent), ventral scales (VEN, number of scales from below mental around the base of the dewlap to anterior border of cloaca), fourth toe lamellae (LAM4, number of 4th toe lamellae, from 1st lamella at the digit's cleft to the most distal lamella), supralabials (SL, posterior end defined by the last enlarged scale that contacts the infralabials at the corner of mouth), infralabials (IL, posterior end defined by the posterior-most enlarged scales that contact the supralabials at the corner of the mouth), ventral scales on the belly (VENB, number of scales posterior to the dewlap to the anterior border of cloaca). Multivariate principal component analysis (PCA) was performed on selected morphometric values: HL, HH, HW, TrL, LAL and CL (Table S2). The variables were selected for the analysis as data

for them was complete. These values were corrected for SVL and were later log-transformed.

## Molecular methods

Molecular data for the gene 16S rRNA generated by *Krishnan (2005)* were downloaded from GenBank, and accession numbers and their collection localities are listed in Table 1. *Gonocephalus pyrius* was chosen as an outgroup for the phylogenetic analysis. The downloaded sequences were aligned in Mega X (*Kumar et al., 2018*) using CLUSTALW (*Thompson, Higgins & Gibson, 1994*) with default settings. Sequences that did not align were excluded from the dataset. The aligned dataset was analysed in a maximum likelihood (ML) framework on the IQ-TREE online portal (*Trifinopoulos et al., 2016*). The optimum sequence evolution model was determined through ModelFinder (*Kalyaanamoorthy et al., 2017*) based on AIC values. The analysis was run with an ultra-fast tree search method (*Hoang et al., 2018*) with 1,000 pseudo-replicates to assess branch support. The resulting tree was visualised and edited with FigTree (*Rambaut, 2012*). Uncorrected p-distances were calculated using Mega X with default settings, and the partial deletion option was chosen to deal with missing data.

# RESULTS

## Molecular analysis

The ML phylogeny was based on 446 bp of mitochondrial 16S rRNA and was rooted with *Gonocephalus pyrius* Harvey, Rech, Riyanto, Kurniawan & Smith, 2021 (Fig. 1, Table 1). The sequence substitution model selected was TPM2u+F+G4. The analysis recovered two clades, clade I, comprising *C. brevicauda*, clade II comprises a basal *Coryphophylax* sp. (*Coryphophylax* sp. 1 Fig. 1) from Car Nicobar Island (the northernmost island in the Nicobar Islands), which is sister to a sub-clade containing representatives from Nicobar Islands (*Coryphophylax maximiliani*) and *Coryphophylax subcristatus* sensu *stricto* (ML bootstrap support 90%). The monophyly of *Coryphophylax maximiliani* + *Coryphophylax subcristatus* sensu *stricto* received low bootstrap support (56%).

*Morphology and nomenclature:* The PCA plot shows the separation of the Nicobar Island population from *C. subcristatus* and *C. brevicauda*, where PC1 + PC2 explain 76.7% (45.7 + 31.1) of the variance (Fig. 2, Table S3). Furthermore, the SVL of mature individuals from the Nicobar Islands is >90 mm, which is much larger than that of *C. subcristatus* and *C. brevicauda*. See detailed comparisons below. The syntypes of *Coryphophylax maximiliani* and *Coryphophylax humei* are morphologically similar and match all the diagnostic characters proposed for both nomina. Genetically, the representatives of samples from their respective type localities show a divergence of 0–4% (Table 2). The divergence is relatively high but may be due to the fragment chosen or sequencing errors. However, a similar range of divergence was observed in representatives of *C. subcristatus*, which suggests that the two nomina from Nicobar Islands are synonymous. As per Article 23, the Principle of Priority, of the *Code* (*International Commission on Zoological Nomenclature, 1999*), the name *Coryphophylax maximiliani* Fitzinger *in Steindachner,*

**Table 1  List of species, samples and their GenBank accession numbers.**

| Current species name | Names in Krishnan (2005) | Code | Locality | 16S |
|---|---|---|---|---|
| C. brevicauda | Coryphophylax sp. 1 | sk04cp4 | Chepo N. Andaman Island | EU502981 |
| C. brevicauda | Coryphophylax sp. 1 | sk04cp11 | Chepo N. Andaman Island | EU502980 |
| C. maximiliani | Nicobar lineage | sk03cs58 | Camorta Island | EU503021 |
| C. maximiliani | Nicobar lineage | sk03cs33 | Katchal Island | EU503007 |
| C. maximiliani | Nicobar lineage | sk03cs34 | Katchal Island | EU503006 |
| C. maximiliani | Nicobar lineage | sk03cs35 | Katchal Island | EU503005 |
| C. maximiliani | Nicobar lineage | sk03cs2 | Kondul Island | EU503004 |
| C. maximiliani | Nicobar lineage | sk03cs4 | Kondul Island | EU503003 |
| C. maximiliani | Nicobar lineage | sk03cs13 | Kondul Island | EU503002 |
| C. maximiliani | Nicobar lineage | sk03cs14 | Kondul Island | EU503001 |
| C. maximiliani | Nicobar lineage | sk03cs5 | Little Nicobar Island | EU502996 |
| C. maximiliani | Nicobar lineage | sk03cs7 | Little Nicobar Island | EU502995 |
| C. maximiliani | Nicobar lineage | sk03cs8 | Little Nicobar Island | EU502994 |
| C. maximiliani | Nicobar lineage | sk03cs6 | Menchal Island | EU502993 |
| C. maximiliani | Nicobar lineage | sk03cs9 | Menchal Island | EU502992 |
| C. maximiliani | Nicobar lineage | sk03cs11 | Menchal Island | EU502991 |
| C. maximiliani | Nicobar lineage | sk03cs16 | Menchal Island | EU502990 |
| C. maximiliani | Nicobar lineage | sk03cs3 | Pulo Milo Island | EU502985 |
| C. maximiliani | Nicobar lineage | sk03cs10 | Pulo Milo Island | EU502984 |
| C. maximiliani | Nicobar lineage | sk03cs15 | Pulo Milo Island | EU502982 |
| C. maximiliani | Nicobar lineage | sk03cs52 | Tillanchong Island | EU502978 |
| C. maximiliani | Nicobar lineage | sk03cs54 | Tillanchong Island | EU502977 |
| C. maximiliani | Nicobar lineage | sk03cs55 | Trinkat Island | EU502976 |
| C. maximiliani | Nicobar lineage | sk03cs56 | Trinkat Island | EU502975 |
| C. subcristatus | Nicobar lineage | sk03cs31 | Chowra Island | EU503020 |
| Coryphophylax sp. 1 | Nicobar lineage ? | sk03cs17 | Car Nicobar | EU503013 |
| Coryphophylax sp. 1 | Nicobar lineage ? | sk03cs19 | Car Nicobar | EU503012 |
| Coryphophylax sp. 1 | Nicobar lineage ? | sk03cs20 | Car Nicobar | EU503011 |
| C. subcristatus | C. subcristatus | sk03cs47 | Chepo N. Andaman Island | EU503018 |
| C. subcristatus | C. subcristatus | sk03cs48 | Chepo N. Andaman Island | EU503017 |
| C. subcristatus | C. subcristatus | sk03cs49 | Chepo N. Andaman Island | EU503016 |
| C. subcristatus | C. subcristatus | sk03cs50 | Chepo N. Andaman Island | EU503015 |
| C. subcristatus | C. subcristatus | sk03cs51 | Chepo N. Andaman Island | EU503014 |
| C. subcristatus | C. subcristatus | sk03cs39 | Havelock Island | EU503010 |
| C. subcristatus | C. subcristatus | sk03cs40 | Havelock Island | EU503009 |
| C. subcristatus | C. subcristatus | sk03cs42 | Havelock Island | EU503008 |
| C. subcristatus | C. subcristatus | sk03cs21 | Little Andaman Island | EU503000 |
| C. subcristatus | C. subcristatus | sk03cs22 | Little Andaman Island | EU502999 |
| C. subcristatus | C. subcristatus | sk03cs23 | Little Andaman Island | EU502998 |
| C. subcristatus | C. subcristatus | sk03cs24 | Little Andaman Island | EU502997 |
| C. subcristatus | C. subcristatus | sk03cs36 | Neil Island | EU502988 |

(Continued)

| Current species name | Names in *Krishnan (2005)* | Code | Locality | 16S |
|---|---|---|---|---|
| *C. subcristatus* | *C. subcristatus* | sk03cs37 | Neil Island | EU502987 |
| *C. subcristatus* | *C. subcristatus* | sk03cs38 | Neil Island | EU502986 |
| *C. subcristatus* | *C. subcristatus* | sk03cs12 | Pulo Milo Island | EU502983 |
| *C. subcristatus* | *C. subcristatus* | SK03CS44 | South Andaman Island | EU502974 |
| *C. subcristatus* | *C. subcristatus* | SK03cs45 | Wandoor South Andaman Island | EU502973 |
| *Gonocephalus pyrius* | | | Lampung, Indonesia | OP070023 |

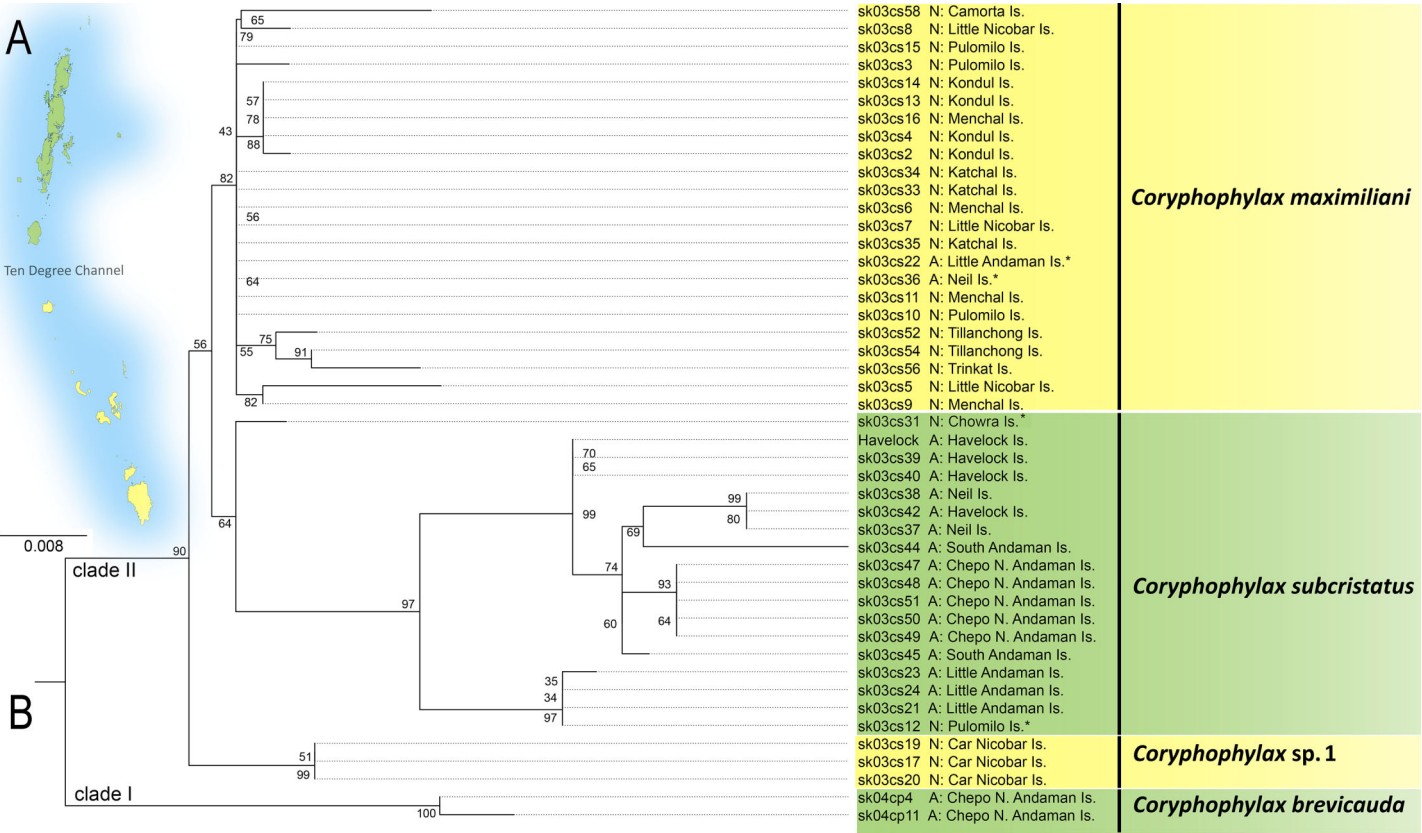

**Figure 1 (A and B) ML phylogeny of members of the genus *Coryphophylax* based on 446 bp of mitochondrial 16S rRNA.** (B) Numbers at nodes represent ML bootstrap support based on 1,000 pseudoreplicates. Map (A) shows the Andaman and Nicobar Islands, and colour on the tips of the tree correspond to color on the map. Samples marked with an asterisk (*) are likely have incorrect collection localities (see Discussion). For the complete tree see Fig. S1.                              

*1867* has precedence over *Stoliczka*'s *(1873) Coryphophylax humei* and is assigned to the large-bodied population of *Coryphophylax* from the Nicobar Islands. The first usage of *Coryphophylax maximiliani* by *Fitzinger, 1861* lacked an associated formal description and was correctly regarded as a nomen nudum by Wermuth (1965). The nomen was made available by Fitzinger *in Steindachner, 1867* (incorrectly given as 1869 by *Wermuth, 1967*),

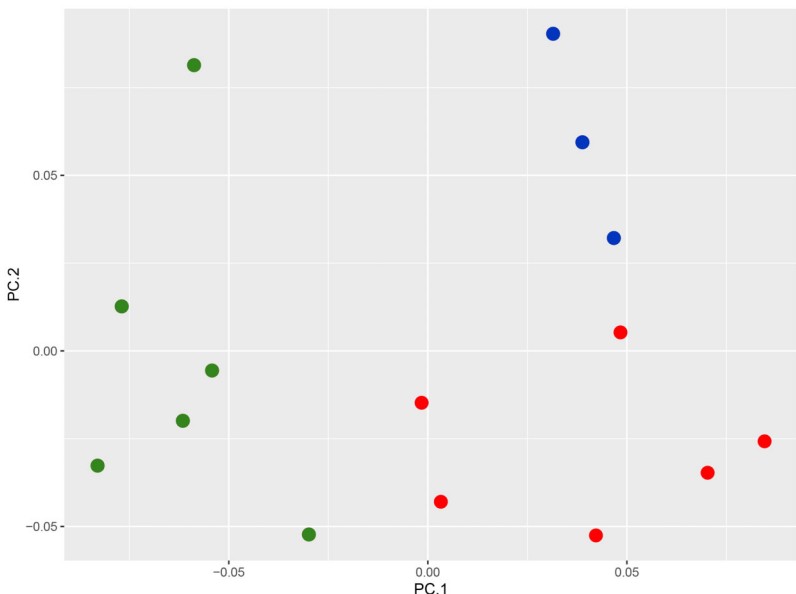

**Figure 2 PCA plot of species of the genus *Coryphophylax*, (green) *C. maximiliani*, (red) *C. subcristatus*, (blue) *C. brevicauda*.**

**Table 2 Un-corrected sequence divergence for *Coryphophylax* species (%). See Table S1 for p-distance for all samples.**

|  | *C. maximiliani* ($n$ = 23) |
|---|---|
| *C. brevicauda* | 5–7 |
| *C. maximiliani* | 0–4 |
| *C. subcristatus* | 3–7 |

where it is accompanied by a detailed description and illustration of one of the syntypes (*Gemel, Gassner & Schweiger, 2019*).

The type specimens of *Coryphophylax maximiliani* are in a fragile condition; only part of the data could be recorded for the largest specimens. The ZSI syntype of *Tiaris humei* is in a much better state of preservation; hence, we here redescribe the species in detail based on all examined material. The members of the genus are morphologically quite similar, and likely, additional species will be described in the near future; it is, therefore, necessary to designate lectotypes for relevant species to stabilise the taxonomy of the group. Hence, lectotypes are here designated for the *Coryphophylax maximiliani* from the series of specimens available at the Natural History Museum, Vienna and for the only traceable specimen of *Tiaris humei* at the Zoological Survey of India, Kolkata.

***Coryphophylax maximiliani* Fitzinger in *Steindachner, 1867***

*Coryphophylax maximiliani* *Fitzinger, 1861*: 387 & 397 (nomen nudum *fide* *Wermuth, 1967*)

*Coryphophylax maximiliani* Fitzinger *in* *Steindachner, 1867*: 30

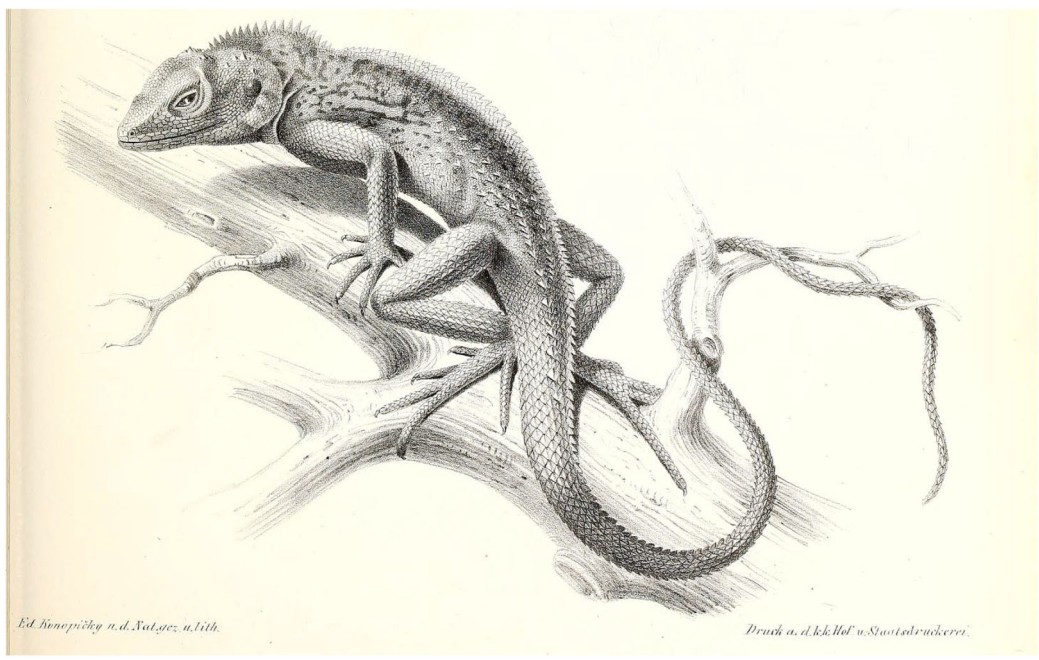

**Figure 3** Illustration of *C. maximiliani* reproduced from Fitzinger in: *Steindachner, 1867* (**Plate II, Fig. 6**). The illustration was altered for visual purposes by removing other illustrations; see Fig. S2 for the unaltered version.                                 

*Tiaris Humei Stoliczka, 1873*: 167

*Goniocephalus humii Boulenger, 1885*: 293; *Boulenger, 1890*: 123

*Goniocephalus subcristatus* (partim) *Smith (1935)*: 163; *Annandale (1904)*: 18; *Sharma (2002)*: 183

*Coryphophylax subcristatus* (partim) *Manthey (2008)*: 99; *Harikrishnan et al. (2012)*: 45

*Coryphophylax maximiliani* [authorship attributed to *Steindachner, 1867*] *Gemel, Gassner & Schweiger, 2019*

*Coryphophylax* Nicobar lineage (*Krishnan, 2005*): 80
    Figures 3–7, Table 3

**Lectotype (here designated):** adult male NHMW 20976:5 from Nicobar Islands (Figs. 3 & 4)

**Paralectotypes (*n* = 8):** juvenile NHMW 20976:1 (SVL 57.9 mm), juvenile NHMW 20976:2 (SVL 62.1 mm), adult NHMW 20976:3 (SVL 93.6 mm, TaL 210.0 mm), juvenile NHMW 20976:4 (SVL 54.4 mm), adult NHMW 20976:6 (SVL 78.0 mm), adult NHMW 20976:7 (SVL 96.0 mm), juvenile NHMW 20976:8 (SVL 58.9 mm), juvenile NHMW 20976:9 (SVL 63.1 mm)

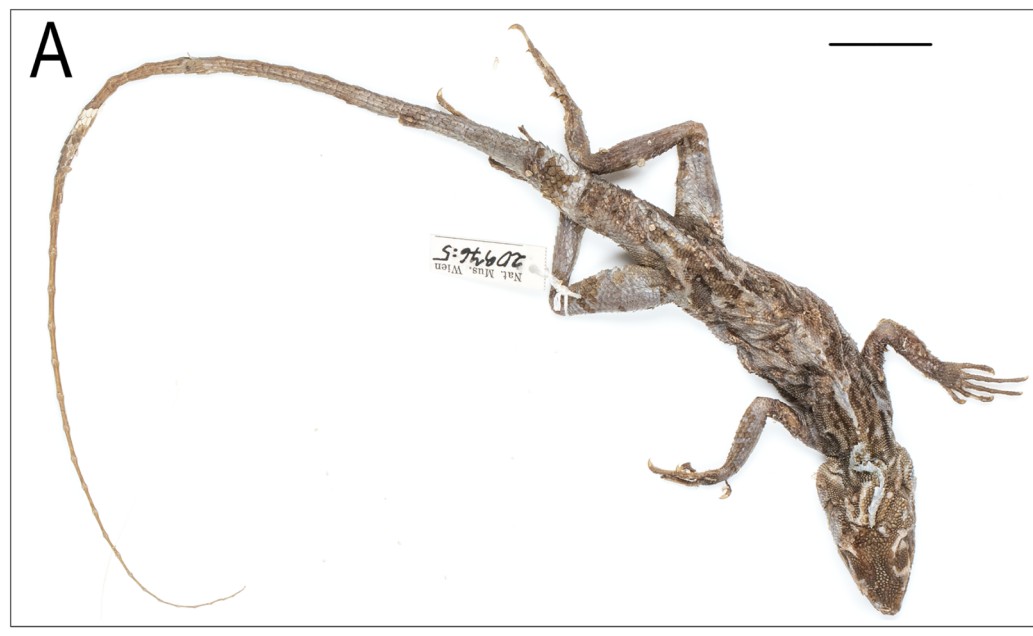

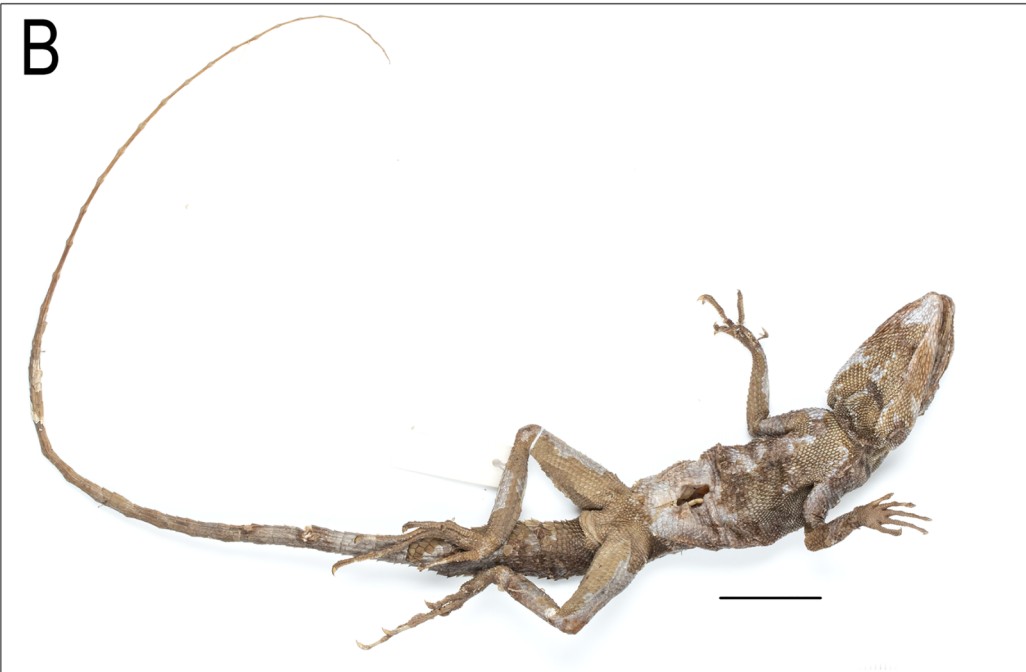

**Figure 4 *Coryphophylax maximiliani* lectotype adult male NHMV 20976:5.** (A) Dorsal view, (B) ventral view. Scale bar 20 mm.

**Additional material examined (*n* = 5):** Lectotype of *Tiaris humei* (here designated), adult male ZSI 5041 from Tillanchong, Nicobar Islands; adult male BNHS 674 from Andaman and Nicobar Islands; adult male ZMB 5854, Nicobars; adult male ZMUC R36998 Camorta, Nicobar Islands; adult male ZMUC R36312 Kondul, Nicobar Islands.

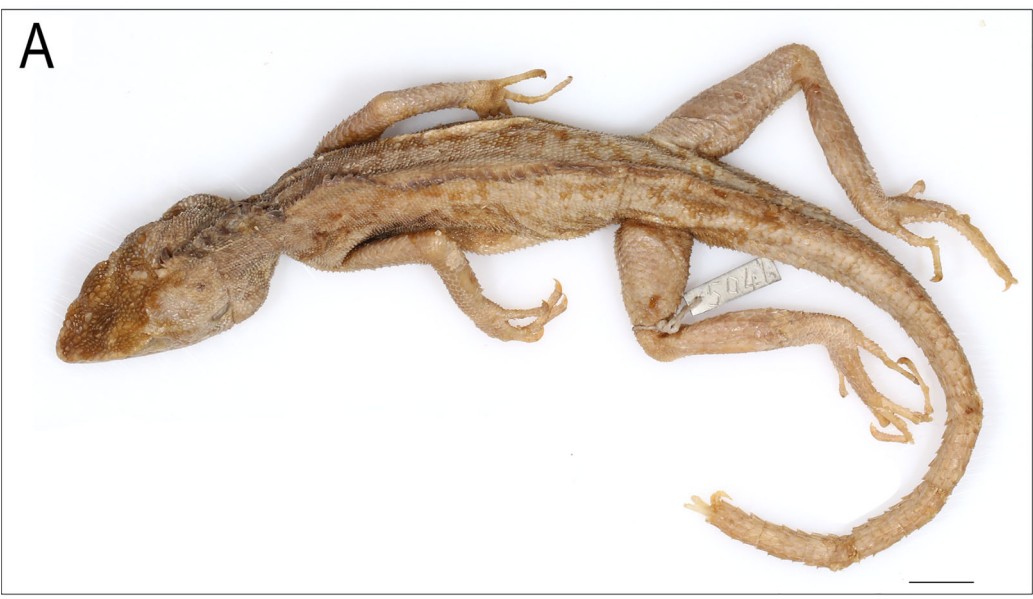

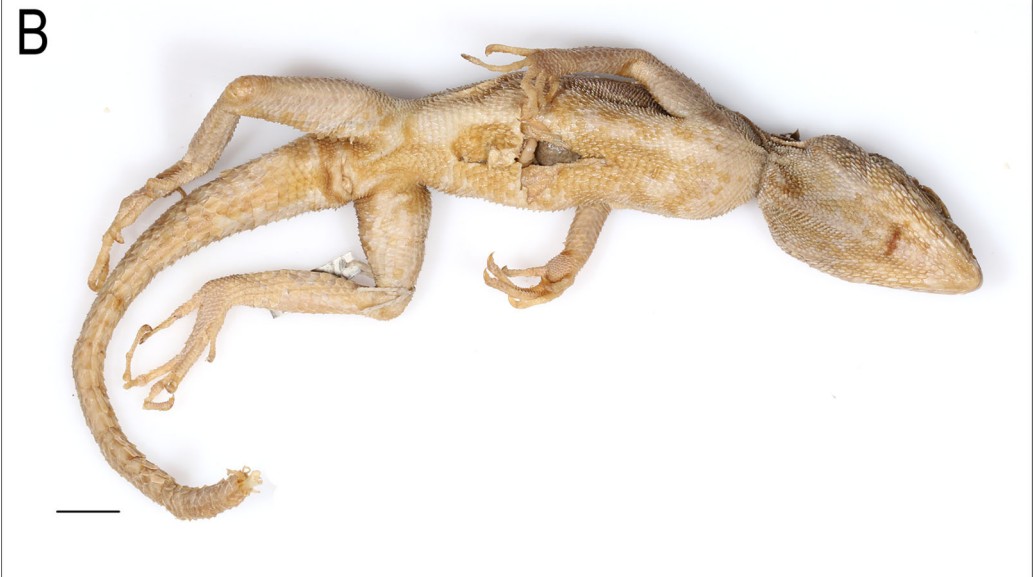

**Figure 5** *Coryphophylax humei* **lectotype male ZSI 5041.** (A) Dorsal view, (B) ventral view. Scale bar 10 mm.                                                           

**Diagnosis:** A large-sized species of the genus *Coryphophylax*, adults measuring 90–108 mm SVL with a TaL/SVL ratio of 2.68–2.73. Midbody scales in 82–85 rows, largely homogeneous in general appearance, intermixed with two fairly well-defined parallel rows of sparse, large tubercle-like scales on the trunk. The nuchal and dorsal crests well developed, composed of an erect flap of skin with slightly larger erected spines forming the apical scale row, which is very distinct in nuchal crest; skin flap differentiated from nuchal to dorsal region, with a small diastema above shoulder dorsal crest continues to 1/4 of the

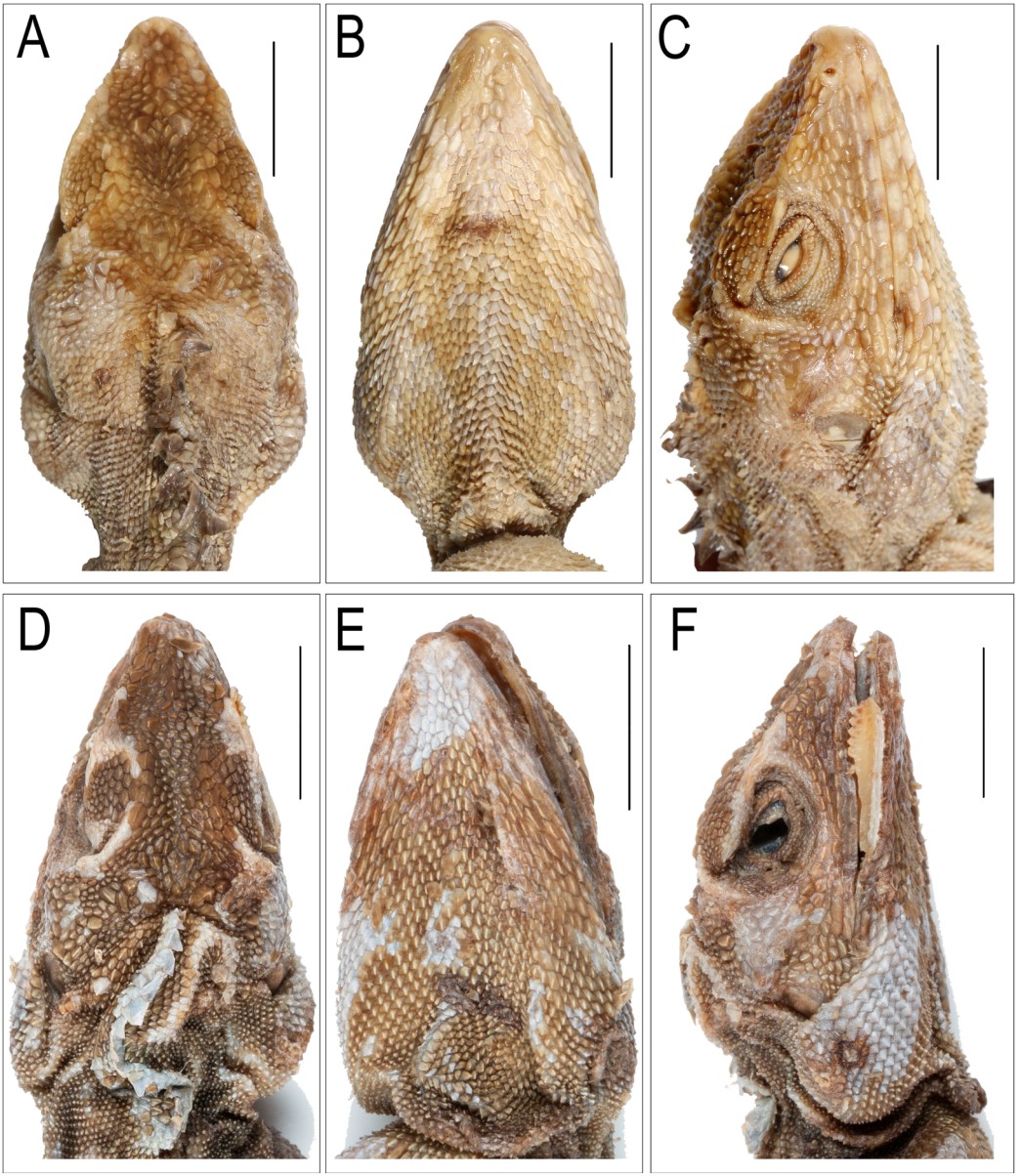

**Figure 6** Images depicting cephalic region of *Coryphophylax humei* lectotype male ZSI 5041. (A–C), *Coryphophylax maximiliani* lectotype adult male NHMV 20976:5 (D–F); (A & D) dorsal view, (B & E) ventral view, (C & F) right lateral view. Scale bar 10 mm.

tail. Dorsal surface of thigh with enlarged keeled scales, one of these largest as seen in members of the genus *Sitana* Cuvier, 1829. 28–31 bi-mucronate lamellae on 4$^{th}$ toe. Male dewlap yellow with black reticulate markings.

**Description of *Coryphophylax maximiliani* based on examined material (Table 3):**

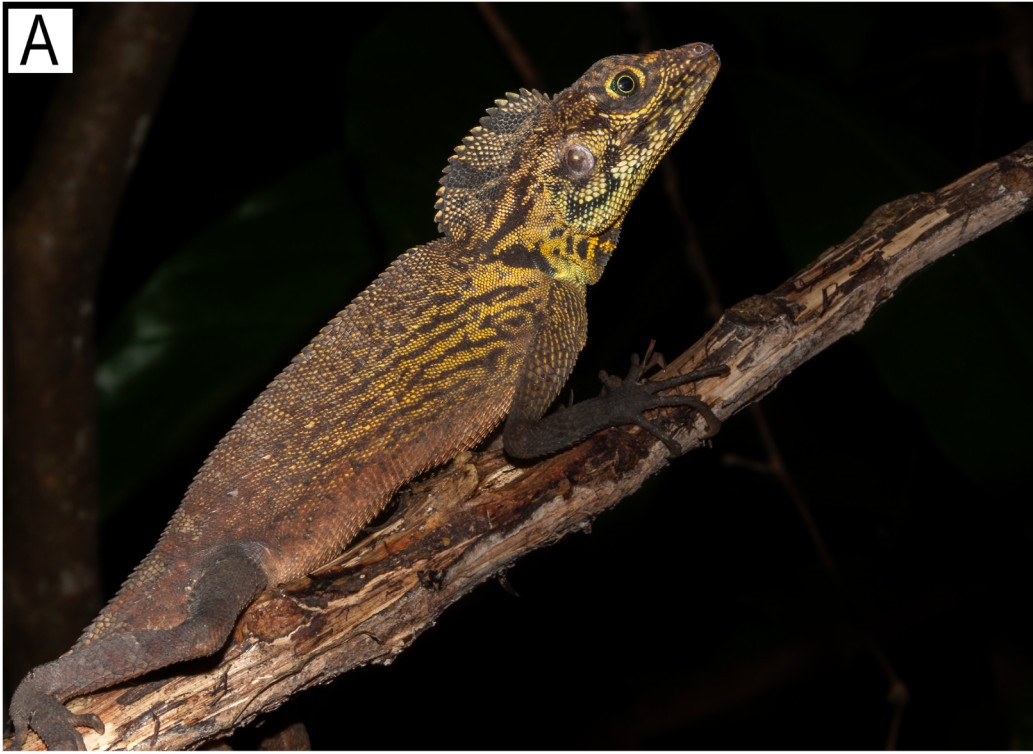

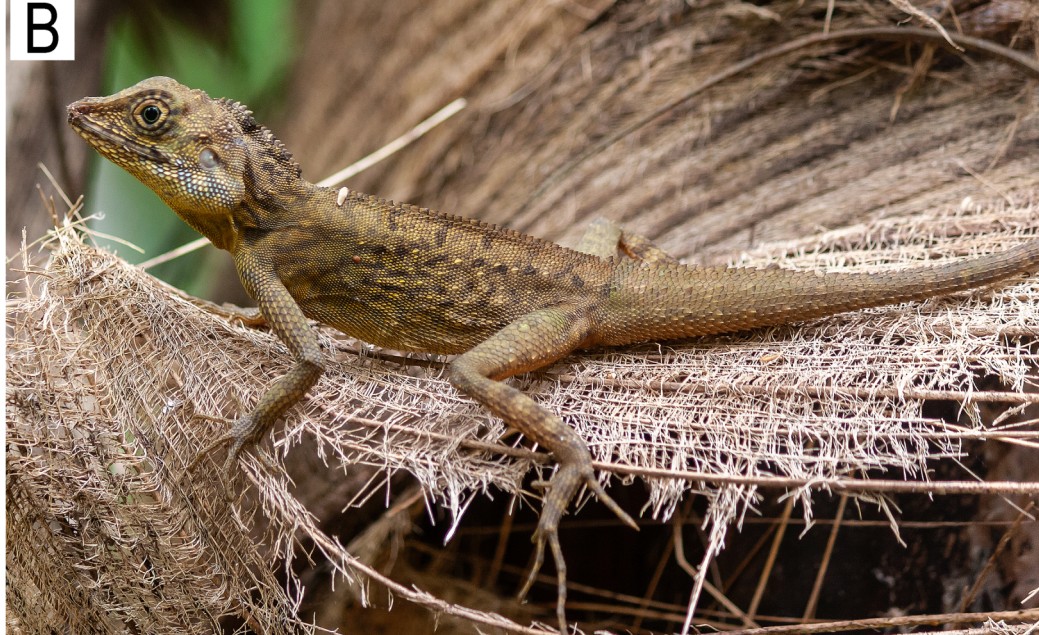

**Figure 7** *Coryphophylax maximiliani* **from Nicobar Islands showing colouration in life.** Photos by S. Harikrishnan.

**Table 3 Morphometric and meristic data for specimens of *Coryphophylax maximiliani*.**

| | Lectotype of *Coryphophylax maximiliani* | Lectotype of *Tiaris humei* | | | |
|---|---|---|---|---|---|
| | NHMW 20976:5 | ZSI 5041 | ZMUC R36998 | ZMUC R36312 | ZMB 5854 |
| Locality | Nicobar Is. | Tillanchong, Nicobar Is. | Comorta, Nicobar Is. | Kondul, Nicobar Is. | Nicobar Is. |
| Sex | male | male | male | male | male |
| SVL | 90 | 108 | 94 | 95.4 | 95 |
| TaL | 246 | — | 180 | — | 210* |
| TaW | 8.4 | 9 | 8.2 | 8.9 | 7.5 |
| HL | 29.7 | 33.1 | 30.3 | 30 | 29.1 |
| HW | 16.1 | 19 | 17.5 | 18.8 | 15.1 |
| HH | 14.4 | 16.8 | 16.3 | 13.9 | 14 |
| SL | 10/10 | 7/ 9 | 9/9 | 9/9 | 7/7 |
| IL | 9/9 | 9/9 | 8/9 | 10 | 9/8 |
| OD | 4.2 | 6.3 | — | — | 4.8 |
| TD | 2.2 | 3.9 | — | — | 3.3 |
| TrL | 40 | 45 | 35.1 | 39 | 41 |
| TrW | 12.7 | 16.5 | — | — | 10.1 |
| LAL | 15.2 | 19.7 | 19.9 | 17.6 | 17.1 |
| CL | 23.9 | 28.9 | 27.2 | 27 | 26 |
| IO | 8.1 | 11.1 | 11.1 | 11.7 | 8.1 |
| IN | 3.6 | 4.4 | 3.8 | — | 3.9 |
| SE | 10.9 | 12.3 | 11.3 | 12.6 | 11.2 |
| ET | 6.8 | 8.3 | — | 6.5 | 6.4 |
| MD | — | 68 | 65 | — | 63 |
| MBS | 85 | 82 | 84 | — | — |
| Right manus | X-10-16-25-14 | 9-12-19-20-11 | 11-18-24-25-15 | 11-18-25-25-15 | 11-16-22-23-13 |
| Left pes | 11-18-26-28-12+ | 10-14-21-28-17 | 11-18-25-28-17 | 12-18-26-30-19 | 11-17-25-31-18 |

**Note:**
Attributes marked with an asterisk (*) indicate missing parts or damage; those with '—' were not recorded as the specimen was shrivelled or damaged.

Adults, SVL 90–108 mm (mean 96.48 ± 6.78). Head relatively long (HL/SVL ratio 0.30–0.33, mean 0.32 ± 0.010), moderately wide (HW/HL ratio 0.52–0.63, mean 0.57 ± 0.040), fairly depressed (HH/HL ratio 0.46–0.54, mean 0.49 ± 0.029), distinct from neck (Fig. 5A). Snout long (SE/HL ratio 0.37–0.42, mean 0.38 ± 0.021), bluntly conical; longer than eye diameter (OD/SE ratio 0.39–0.51, mean 0.44 ± 0.065) (Fig. 6C). Eye large (OD/HL ratio 0.14–0.19, mean 0.17 ± 0.025); pupil round, eyelids covered with small pentagonal and hexagonal scales, supraciliaries short. Snout obtusely pointed when viewed dorsally, rostral much wider than deep, bordered posteriorly by first supralabial, prenasal and dorsally by four small scales. Canthus rostralis and supraciliary edge sharp consisting of 12–14 scales, some of these are large tuberculate. Nostrils positioned centrally in a large, undivided nasal plate, bordered by 9–10 scales, including one prenasal, four postnasals and

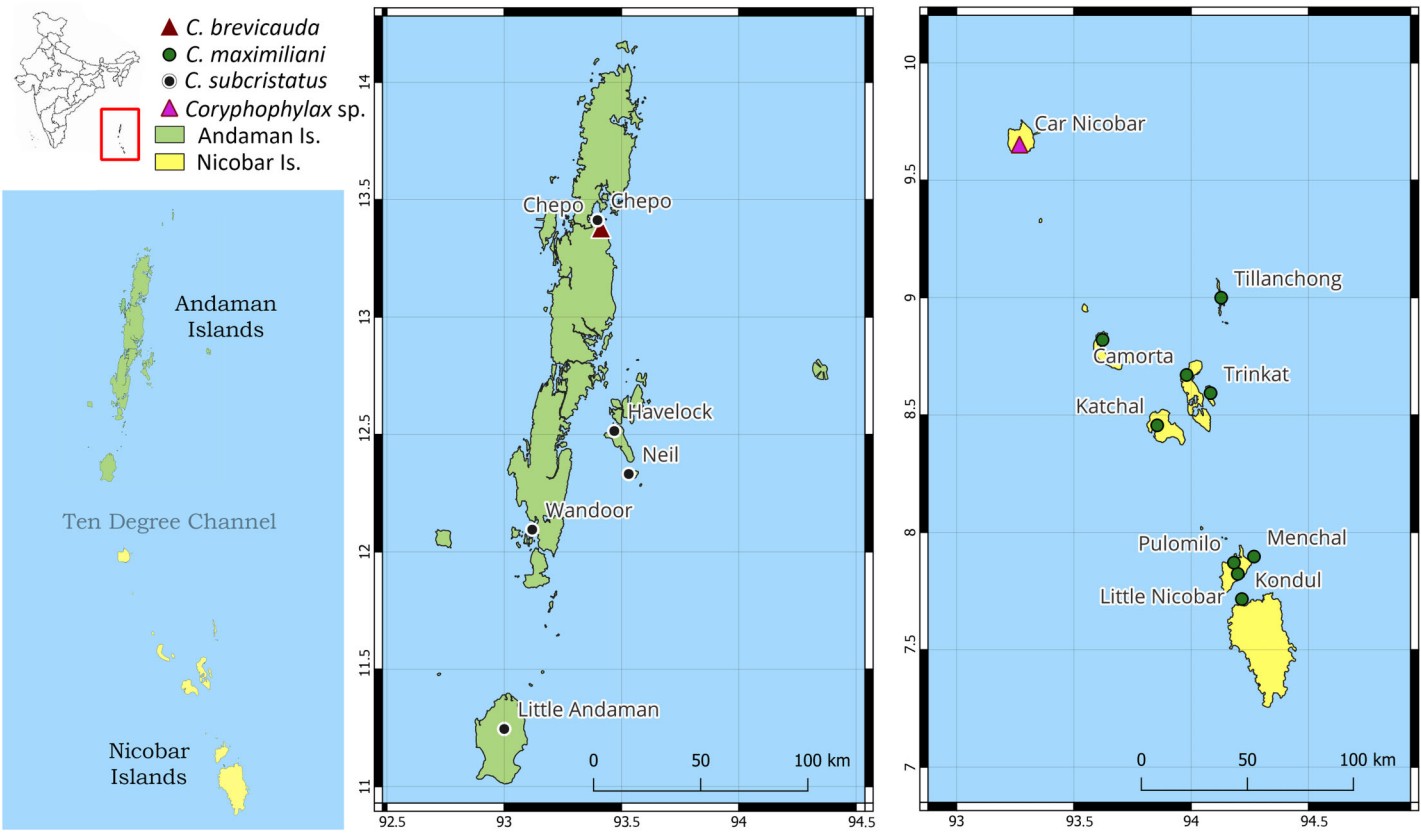

**Figure 8** **Map of Andaman and Nicobar Islands showing collection localities for molecular data.** The inset map of India (top left) highlights the region of interest by the red rectangle, and the map below shows the Andaman and Nicobar Islands and the Ten Degree Channel.

two supranasal, and in contact with rostral. Supralabials 7–10 rectangular, weakly keeled, bordered above by a single row of slightly smaller, rectangular, keeled scales. Loreal region concave, scales of the loreal region heterogeneous in size, raised not flat, keeled, some roughly hexagonal. Scales on postorbital and temporal region heterogeneous, imbricate, strongly keeled, and directed posteriorly and dorsally. Orbital scales small but not granular. Tympanum naked. Four to five large, strongly keeled, tuberculate scales running from the posterior part of orbit to the supra-tympanic region. A large tuberculate scale on the nape on either side of the dorsal crest; a second subequal tuberculate scale in the supratympanic region and postocular region. Canthals enlarged, overlapping, becoming slightly smaller along subimbricate supraciliaries, protruding slightly laterally on supraorbital ridge. Scales on dorsal surface of snout, forehead, interorbital, and occipital region heterogeneous in size, and shape; mostly elongate, imbricate, strongly keeled longitudinally; those on snout smaller, rhomboidal, those on the supra-occiput largest. Parietal plate without pineal eye, the plate slightly larger than adjacent scales. Mental shield narrower than rostral; gular scales keeled (Fig. 6B). Infralabials 8–10. Nuchal absent, and dorsal crest present,

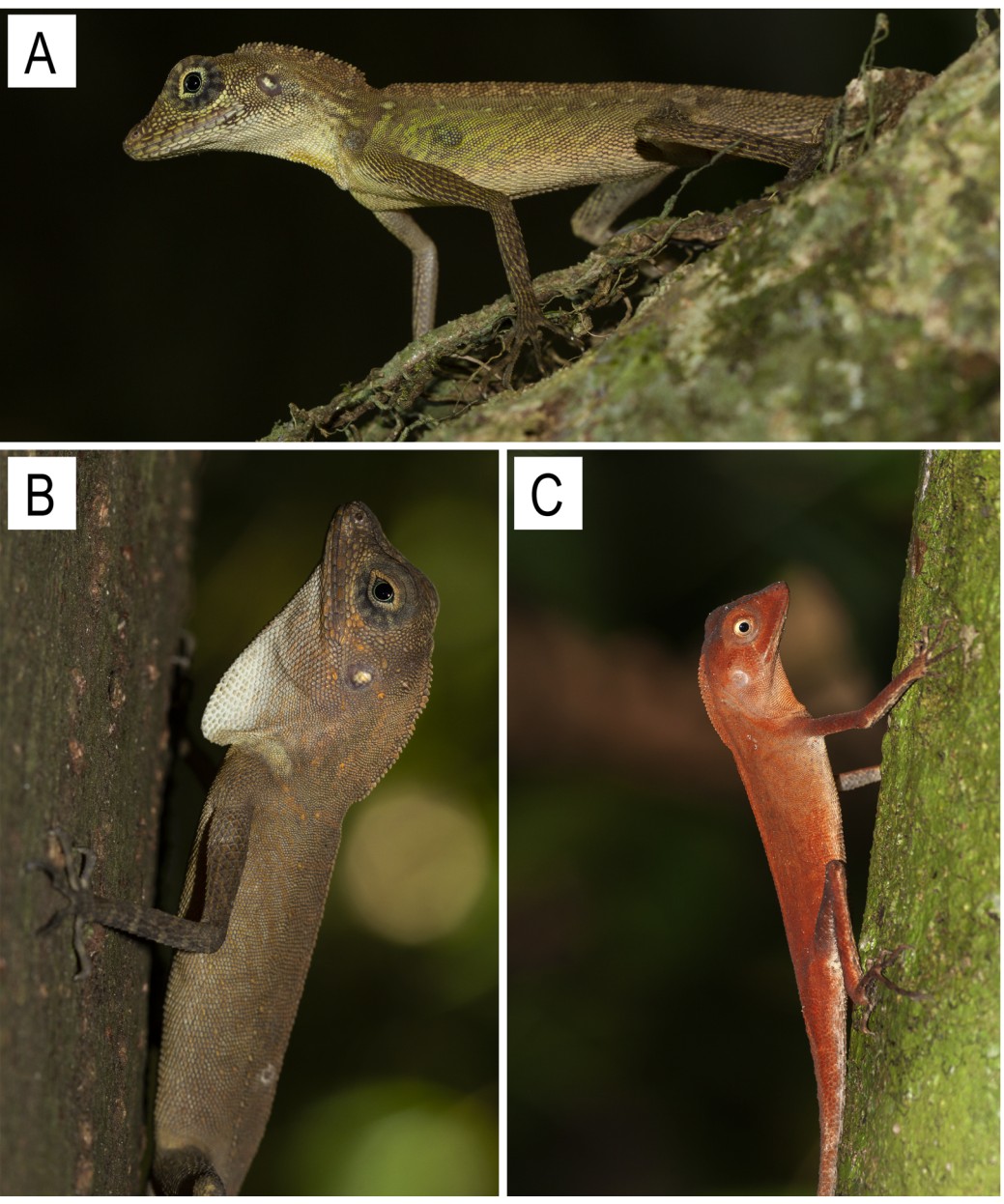

**Figure 9** Images of *Coryphophylax* species (A & B) *C. subcristatus*, photo by Zeeshan A. Mirza; (C) *Coryphophylax brevicauda*, photo by S. Harikrishnan.

composed of low thorn-like scales. Scales on nuchal region smaller, less than half the size of those on interorbital region, imbricate, strongly keeled. Dorsal crest comprising 57–58 raised, spike-like scales in a row on a raised flap of skin running from the posterior part of the head to the level of the vent. The crest spines high and erect with a disatema, and relatively short spine-like scales run along the vertebral column to the anterior 1/4$^{th}$ of the tail. Body slender (Fig. 5A), 80–85 rows of scales around midbody, of these 16–18 rows across the belly are slightly larger than those on the dorsum (Fig. 5B); from neck to pectoral

region scales largely homogenous, feebly keeled, intermixed with large strongly keeled scales. twice to thrice as big, scales on the trunk slightly larger than those on neck, imbricate, pointed, keeled, and directed posterodorsally forming regularly arranged longitudinal rows; ventral scales strongly keeled arranged in 16–18 rows, oriented backwards, subimbricate, heteroogenous in size; no precloacal or femoral pores (Fig. 5B). Number of scales MD 63–68. Distinct fold at the shoulder present. Fore and hind limbs relatively slender, tibia short (CL/SVL ratio 0.27–0.29, mean 0.28 ± 0.010); digits moderately long, ending in strong, elongate, slightly recurved claws; inter-digital webbing absent; subdigital lamellae entire, bi-mucronate, 20–25 subdigital lamellae on finger IV of manus and 28–31 on finger IV of pes; relative length of fingers and toes 4 > 3 > 5 > 2 > 1. Fore and hind limbs covered above and below with regularly arranged, enlarged, pointed, strongly keeled scales. Dorsal and dorso-lateral scales on the posterior part of the thigh enlarged, one of these scales largest and projecting as seen in members of the genus *Sitana* Cuvier, 1829. Tail entire; tail base swollen; tail uniformly covered with similar sized, keeled, weakly pointed, regularly arranged, posteriorly directed imbricate scales, no enlarged median subcaudal row; erect.

**Distribution:** Based on examined specimens and molecular data, the species appears to be distributed on the following islands in the Nicobar group of islands: Little Nicobar Is., Camorta Is., Kondul Is., Menchal Is., Katchal Is., Tillanchong Is., Trinkat Is. See Figs. 1, 7 & 8 for specific localities. Likely distributed throughout the Nicobar Islands except for Great Nicobar Island.

**Comparisons:** *Coryphophylax maximiliani* differs from its two congeners as follows: SVL 90–108 mm (*vs.* 53–85 mm in *C. subcristatus* (Fig. 9A & 9B), 42–63 mm in *C. brevicauda* Fig. 9C); TaL/SVL ratio 2.68–2.73 (*vs.* 2.04–2.51 in *C. subcristatus*, 1.69–2.03 in *C. brevicauda*); 82–85 scale rows round the body (*vs.* 85–100 in *C. subcristatus*, 110–121 in *C. brevicauda*); dorsal and nuchal crest comprising a skin flap expansion with large erect spine-like apical scales on the nape, with a diastema followed by relatively short spines running along the vertebral column extending to 1/4 of the tail (*vs.* dorsal and nuchal crest well developed, low with short spine-like scales running continuously from nape to mid-trunk in *C. subcristatus*, dorsal crest low and lacks large erect spine-like scales in *C. brevicauda*) (Fig. 8); dewlap colour yellow with black reticulate markings (*vs.* dewlap yellow or white without any markings in *C. subcristatus*, orange-red in *C. brevicauda*).

## DISCUSSION AND CONCLUSIONS

The phylogenetic affinities of the genus *Coryphophylax* remain poorly studied. The only study that assessed the phylogenetic placement of the genus based on molecular data was by *Pal et al. (2018)*. That study recovered *Coryphophylax* as the sister taxon to *Bronchocela*, a relationship contingent on the 16S rRNA sequence of a single species. The present work also suffers from the limited molecular data currently available (sequences of a fragment of 16S rRNA only). However, it is of merit to present a snapshot of the diversity of the lizards across these islands and confirm that 'C. subcristatus' comprises multiple species. Nevertheless, the monophyly of *Coryphophylax* must be tested with nuclear genes. The

genus *Coryphophylax* is endemic to the islands, and its members are distributed throughout both the Andaman and Nicobar groups, except for Great Nicobar Island. Nearly all other reptile groups are either distributed in the Andamans or Nicobars or may have narrow distribution, being restricted to a few islands (*Das, 1999*; *Chandramouli et al., 2023*; *Vijayakumar & David, 2006*). The abundant and widespread nature of *Coryphophylax* across the islands make them an ideal model species for studying the patterns of colonization and diversification across the Andaman and Nicobar Islands.

The results of the present study align with those of *Krishnan (2005)*, who, based on a large sample size of specimens for morphological data, considered the Nicobar population distinct. Although the present study lacks the extensive sampling conducted by *Krishnan (2005)*, it arrives at an identical conclusion. This study relied entirely on museum material, and the absence of fresh specimens means that the morphological and molecular data from corresponding specimens are lacking. The molecular data, a short fragment of mitochondrial 16S, are insufficient for resolving phylogenetic relationships. The findings primarily stem from the morphology of museum samples and a few representatives of samples collected by *Krishnan (2005)* housed in the BNHS collection. A significant pitfall of the study is that the data for morphology and sequence were derived from different specimens.

*Tiaris humei* was described based on two specimens from Tillanchong, Nicobar Islands (*Stoliczka, 1873*). The male specimen housed in the collection of the Zoological Survey of India (ZSI 5041) is here designated as the lectotype. The second syntype has been purported to be in the collection of the Natural History Museum, London (NHMUK) (*Das, Dattagupta & Gayen, 1998*). A search through the online portal of the database of NHMUK identified six specimens of *Gonyocephalus subcristatus* from the Nicobar Islands. Two of these specimens bear the catalogue number NHMUK 1934.11.2.30–31 and likely were lodged in the collection in the 1900s, long after the description of *Tiaris humei*. The other four specimens, NHMUK 1868.7.8.12–14 and NHMUK 1874.4.29.1229, appear to have been catalogued around the time of the description of the species. Of the four specimens, three of these NHMUK 1868.7.8.12–14, were donated by J. T. Reinhardt and NHMUK 1874.4.29.1229 was donated by R. H. Beddome and all these specimens are listed as non-type specimens. Further, *Boulenger (1890)*, in his compilation on the reptiles and amphibians of the Indian region, stated in the description of '*Goniocephalus humii*' that 'I have not seen examples of this species'. This suggests that even as early as 1890, the second syntype of *Tiaris humei* was not at the British Museum of Natural History (now NHMUK). Therefore, the whereabouts of the second syntype remain unknown, and the specimen may be lost.

Molecular data suggest that *C. subcristatus* sensu *stricto* is restricted to the Andaman Islands. However, a few samples likely have incorrect localities and require further confirmation. These are marked with '\*' in Fig. 1. Sequences generated by *Krishnan (2005)* bear field numbers, which are in a series. The samples from Little Andaman Island are part of the series sk03cs21, sk03cs23 and sk03cs24. Another sequence, sk03cs12, placed in the same clade, is ostensibly from Pulomilo Is. (Nicobar Is.). This number was apparently

transposed with 'sk03cs22', a sample embedded in the Nicobar clade but bearing the locality 'Little Andaman Is.'. Therefore, we propose removing it from the Nicobar Islands lizard fauna pending confirmation of its occurrence. The populations south of the Ten Degree Channel represent *C. maximiliani* and an undescribed species. The species *C. maximiliani* and *C. subcristatus* show considerable divergence across representative samples (Table 2). This may be attributed to the lack of gene flow between isolated populations on the different island groups. These isolated genetically divergent populations from Andaman Is. and Nicobar Is. correspond to *C. subcristatus* and *C. maximiliani*, respectively.

The revalidation of the nomen from the Nicobar Islands is not surprising, as former workers have suggested that *C. subcristatus* is a species complex (*Das, 1999*; *Krishnan, 2005*; *Vijayakumar & David, 2006*; *Harikrishnan et al., 2012*). The Ten Degree Channel has been proposed as a barrier to gene flow (*Das, 1999*) that would result in reciprocal monophyly of taxa north and south of the channel. However, results from the molecular phylogeny (Fig. 1) did not recover this pattern. The findings hint at several trans-marine dispersal events across this barrier (*Krishnan, 2005*), likely during periods of low sea level (*Gornitz, Lebedeff & Hansen, 1982*; *Voris, 2000*). The aforementioned hypothesis must be tested by employing multiple molecular markers to elucidate the temporal diversification patterns of *Coryphophylax* from its sister taxa and species within the genus. Generating data to test the hypothesis is out of the scope of the present work, largely due to the difficulty of procuring permissions from the Forest Department to conduct research on the islands. The current study also indicates the presence of additional undescribed species and lays a foundation for further studies on this genus.

## ACKNOWLEDGEMENTS

The study was primarily based on museum material would not have been possible without the support of the following directors/curators who helped with access to specimens, data, images, and data: Rahul Khot (BNHS), Peter Rask Møller & Daniel Klingberg Johansson (ZMUC), Silke Schweiger and Georg Gassner (NHMW), Mark-Oliver Rödel and Frank Tillack (ZMB), Dhriti Banerjee and Pratyush P. Mohapatra (ZSIK) and Patrick Campbell (NHMUK). Saunak Pal thanks the entire natural history collection department and the Director of BNHS for permission to study specimens and for their constant support. We thank S. Harikrishnan, Nitya P. Mohanty, and Shashank Dalvi for providing live images of *Coryphophylax* and discussing the distribution of the genus. Special thanks to Ht. Decemson, two anonymous reviewers and editor Viktor Brygadyrenko for their comments that helped improve the manuscript.

## INSTITUTIONAL ACRONYMS

| | |
|---|---|
| **BNHS** | Bombay Natural History Society, Mumbai (India) |
| **NHMW** | Natural History Museum, Vienna (Austria) |
| **ZMB** | Museum für Naturkunde, Berlin (Germany) |
| **ZMUC** | Zoological Museum, University of Copenhagen, Copenhagen (Denmark) |
| **ZSI** | Zoological Survey of India, Kolkata (India) |

### Funding
The Max Planck Society's IMPRS 'From Molecules to Organisms' program supported the work conducted by Zeeshan A. Mirza. The funders had no role in study design, data collection and analysis, decision to publish, or preparation of the manuscript.

### Grant Disclosures
The following grant information was disclosed by the authors:
The Max Planck Society's IMPRS 'From Molecules to Organisms' Program.

### Competing Interests
Tejas Thackeray and Harshil Patel are employed by Thackeray Wildlife Foundation

### Author Contributions
- Zeeshan A. Mirza conceived and designed the experiments, performed the experiments, analyzed the data, prepared figures and/or tables, authored or reviewed drafts of the article, and approved the final draft.
- Saunak Pal conceived and designed the experiments, performed the experiments, analyzed the data, prepared figures and/or tables, authored or reviewed drafts of the article, and approved the final draft.
- Tejas Thackeray conceived and designed the experiments, prepared figures and/or tables, funding, and approved the final draft.
- Harshil Patel conceived and designed the experiments, performed the experiments, analyzed the data, prepared figures and/or tables, authored or reviewed drafts of the article, and approved the final draft.
- Aaron M. Bauer conceived and designed the experiments, analyzed the data, authored or reviewed drafts of the article, and approved the final draft.

### Animal Ethics
The following information was supplied relating to ethical approvals (*i.e.*, approving body and any reference numbers):

The study was based on museum specimens, and no live animal was captured or collected, and hence, no ethical clearance was required for this study.

### Data Availability
Raw data is available in the Supplemental FIles.

### Supplemental Information
Supplemental information for this article can be found online at http://dx.doi.org/10.7717/peerj.19841#supplemental-information.

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
