# Peer review of "Taxonomic status of Coryphophylax maximiliani Fitzinger in: Steindachner, 1867 with notes on Coryphophylax subcristatus (Blyth, “1860” 1861)"

_PeerJ, doi:10.7717/peerj.19841_

## Round 0.1 · original submission · Major Revisions

Dear authors, I kindly ask you to make significant additions to the manuscript in accordance with the reviewers' comments. I hope that your detailed responses to each of the comments will allow the reviewers to approve this publication.

·

Basic reporting

No comment.

Experimental design

No comment.

Validity of the findings

No comment.

Additional comments

I appreciated the authors for their contributions and output for this interesting work based on the taxonomic status to resolve the species Coryphophylax maximiliani with note on C. subcristatus from the Andaman and Nicobar Islands, India.

Regards and best wishes!

Reviewer 2 ·

Basic reporting

No comment.

Experimental design

The authors attempt to validate specific status of the Nicobarese population of Coryphophylax subcristatus based on the morphology of museum preserved specimens and previously published mitochondrial sequence data. The study provides valuable insights on the morphological differences between the two populations of Coryphophylax. However, I find the evidence insufficient for the taxonomic change they propose.

Validity of the findings

I’ve listed the major concerns I find with this study below. All minor comments are listed inline with the attached manuscript.

The Ten-degree channel as a biogeographic barrier:
The authors rightly cite studies that impute this channel as a barrier separating the Sundaic elements in the Nicobars from the species with Burmese affinities in the Andamans. However, Coryphophylax is likely to be Sundaic, given its phylogenetic affinities with either Gonocephalus or Bronchocela. It is therefore also likely that they dispersed from the Nicobars to the Andamans in the recent past, given how similar the populations are (morphologically and genetically). Further, the phylogeny does not support the monophyly of the Andamans versus the Nicobarese Coryphophylax. Therefore, invoking the Ten-degree channel as a biogeographic barrier for this specific case is excessive, and incorrectly alludes to distinct biogeographic histories (Sundaic vs. Burmese) for the two populations of Coryphophylax.

The 16S mitochondrial phylogeny:
The phylogeny presents insufficient evidence for speciation to have occurred between the Andamans and Nicobars populations. C. subcristatus and C. maximiliani are not even reciprocally monophyletic, which is the least you would expect when testing the phylogenetic species concept. Further, other gene trees may recover different relationships between the two groups. The mitochondrial genome fixes > 4 times faster than the nuclear genome within subpopulations, and therefore, high divergence between them is not surprising. Moreover, although these populations are presently isolated, the evidence presented solely based on the 16S phylogeny and p-distances cannot confirm independent evolution of these populations. For instance, there may have been recent, homogenizing gene flow between these populations that has been obscured by mitochondrial divergence (presented in this study).

Specimens used for morphological and genetic analyses:
The major issue I see with this study is that there is no way of mapping the individuals used for the morphological analyses with their 16S sequences. These completely independent sets of individuals are presumptively mapped in the study based on the archipelago they originate from. This is problematic since there is no phenotype information for the sequences used and vice-versa.

Taxonomic changes:
The proposed taxonomic change of elevating the Nicobarese Coryphophylax to species status is therefore not supported by sufficient evidence in this study, which is constrained by data availability. The authors write in their discussion that their hypothesis needs to be validated by employing multiple molecular markers (sic), transferring this burden to future workers. But, based on morphological variation and geographic isolation of maximiliani in the Nicobar Islands, and the lack of evidence for speciation between the Andamans and Nicobarese C. subcristatus, I believe there is reason to elevate maximiliani to a subspecific rank.

Major Line comments:

L114: Why were these variables chosen over all others? How many individuals were used to perform the PCA? Did you use bootstrap/cross validation methods to determine if your sample size (given the number of variables) is large enough to conduct such an analysis?
L122: Why was Bronchocela not chosen as outgroup, given there are plenty 16S sequences available for the genus? I don’t expect too many changes within ingroup relationships, but I suggest the authors rebuild the phylogeny using Bronchocela as outgroup.
L137: Premature to assume that the ‘basal’ linage represents a new species without further data. Please rephrase appropriately, for example: ‘Individuals from Car Nicobar cluster together in the phylogeny with high bootstrap support and are sister to the clade comprising C. maximiliani and C. subcristatus.”
L141: The low to medium branch support for the C. subcristatus and C. maximiliani clades must be mentioned here.
L141: The sample from Chowra islands (south of the 10-degree channel) branches with individuals from the Andamans. Similarly, the samples from Little Andaman and Neil island (north of the 10-degree channel) branch with individuals from the Nicobars. These results clearly imply that the 10-degree channel is not likely to have caused cladogenesis in Coryphophylax. Based on the reconstructed 16S phylogeny, individuals from the Andamans and Nicobars are not reciprocally monophyletic. This has to be mentioned here in the results.
L144: In the PCA plot, does the ‘Nicobar’ population include the individuals from Andamans that branch with it in the phylogeny? Similarly, does the ‘Andaman’ population include the individual from Chowra islands?
L145: What about the SVLs of individuals from Chowra, Neil and Little Andaman islands? Do they conform to C. subcristatus or C. maximiliani ?
L150: Genetic cutoffs cannot be used to split or merge species.
L167: If the lectotype is the ZSI specimen, mention that here.
L182: Several paralectotypes have much smaller SVLs. If these are juvenile/sub-adult individuals, they need to be mentioned here. I suggest mentioning if each specimen in the type series is male/female, adult/juvenile.
L253: What about the samples from Little Andamans and Neil islands that branch with C. maximiliani in the 16S phylogeny?
L257: Harikrishnan et al. (2012) pp 38., say C. subcristatus have an SVL up to 90 mm. Please reconfirm all your morpho/meristics numbers with published literature.
L274: This study does not confirm independent evolution of C. subcristatus and C. maximiliani. At best, it confirms there are two phenotypes within C. subcristatus.
L302-309: You argue that any collection locations that do not conform to the cladistic relationships you seek, are mislabelled. Is there any evidence to say these individuals were indeed mislabelled? Has S. Krishnan, the original collector of these specimens, confirmed such an error?
L312: I agree these populations are presently isolated. However, gene-flow in the very recent past could have homogenized these populations, which could reflect even today in their nuclear genomes. Therefore, there is no molecular evidence to confirm or refute the theory that these populations today represent independently evolving lineages.
L323-324: Indeed. This hypothesis should be tested with several nuclear markers, if not genome-wide markers. I understand this study is restricted by the usage of the mitochondrial 16S marker. However, I disagree with making taxonomic changes you propose with restricted data and insufficient evidence.
L325-326: I disagree that this is out of scope. You intend to make taxonomic changes to this group. So how does acquiring more data/evidence that support (or refute) your hypothesis become ‘out of scope’ for your study?

Additional comments

All minor comments are inline with the attached manuscript.

Annotated reviews are not available for download in order to protect the identity of reviewers who chose to remain anonymous.

Reviewer 3 ·

Basic reporting

This paper reassessed the taxonomy of Coryphophylax subcristatus based on available museum material and 16S sequences retrieved from GenBank. As the authors rightly state, the molecular information is limited, and the phylogeny is not well resolved. Nevertheless, some valuable new insights can be gained from their analyses, and there is apparently no guarantee to get more molecular data in a near future due to permit restrictions. The authors resurrected C. maximiliani and highlighted the occurrence of an undescribed species on Car Nicobar Island (not described in the present work).
The genetic data come from GenBank and were apparently deposited in the framework of an unpublished MSc thesis [Krishnan, Shreyas. "Phylogenetic status and systematics of the agamid Coryphophylax Blyth, 1861 (Reptilia: Squamata)" (2006), not cited in the paper]. I found myself wondering why these data have never been published before, but also if the authors had access to that thesis, and, if yes, why the results are not discussed.

Experimental design

no comment

Validity of the findings

no comment

Additional comments

I have a few minor comments, suggestions, and questions for the Editor and authors to consider:

Title: “Taxonomic status of the Coryphophylax maximiliani Fitzinger in: Steindachner, 1867 with notes on Coryphophylax subcristatus (Blyth, “1860” 1861)”
Why “the” Coryphophylax maximiliani?

Line 16: “represented by two putative species”
Why “putative”? These taxa are recognised species

Lines 23-24: “The members of the genus Coryphophylax are abundant and widespread across the islands”
I am not sure what you mean by “members”. With your study there are now 3 described species in that genus, which can’t be qualified as “abundant”. Do you mean that populations of each species are widespread and that specimens occur in high density? Please clarify

Line 35: “The members of the genus are diurnal and abundant…”
Perhaps replace by “Coryphophylax species are diurnal with large population sizes”?

Lines 82-83: “The study was based on museum material, and no live individuals were captured or collected for this study”.
Perhaps replace by “The study was exclusively based on museum material”?

Line 120-121:“Molecular data for the gene 16S rRNA generated by Shreyas Krishnan was downloaded from GenBank”
“were downloaded” not “was downloaded”
Also, these data seem to come from a MSc thesis, right? Why not citing that work instead of only providing the name of the colleague who generated them?

Line 122: replace “choosn” by “chosen”

Lines 123-124: Did you exclude regions that were ambiguously aligned?

Line 125: Which version of IQ-TREE did you use?
I am not 100% sure, but I think that the reference you provide applies to the software itself and that “Trifinopoulos et al. 2016” (see below) is the correct reference when you use the web-server.
Trifinopoulos J, Nguyen LT, von Haeseler A et al. W-IQ-TREE: a fast online phylogenetic tool for maximum likelihood analysis. Nucleic Acids Research 2016;44:W232–235.

Line 129: replace “Un-corrected p-distance was” with “Uncorrected p-distances were”

Lines 135-136: “The analysis recovered two clades, clade I, comprising C. brevicauda, which was sister to clade II”
Well, if you have two clades, they are by default sister to each other. Perhaps replace with “The analysis recovered two clades, one (clade I) corresponding to C. brevicauda”?
What is the support for these two clades? I do not find it in your text, and it is not provided in Fig. 1 nor in the SI

Line 147: replace “brevacauda” with “brevicauda”

Line 149: I would remove “too” after genetically

Line 150: I would use “suggests” instead of “corroborates”.
Note that 4% divergence in such a short fragment of 16S is not negligible. Since the morphology seems conserved, this might be due to isolation by distance (but as far as I understand you have only one specimen from Tillingchang, right?). However, I must say that I find quite difficult for the reader to compare the genetic distances you provide in Table S1 and get a clear picture of synonyms and their exact distributions. How is the reader supposed to know which samples are “humei” and maximiliani “from their respective type localities” (L149-150) in Table S1 for instance? As far as I understand, you report the type locality of maximiliani as Nicobar Islands, so basically the whole archipelago. Since you mention this genetic divergence of up to 4%, I was interested to compare genetic divergences among islands. But to do that I have to check Figure 1, take note of the sample number/locality, then go in Table S1 (where no localities are provided), then try to see on Figure 8 where the sample is from. I gave up after a few minutes since I have other things to do. Could you please provide this information in a more intelligible way?
I am not familiar with that region, but “Tillingchang” is the name you refer to in the Introduction (as a correction of “Tillinchang”), but elsewhere in the MS and in the figures you wrote “Tillachong”.

Line 257: “63 mm” not “63mm”

Line 268: “The phylogenetic affinity of the genus Coryphophylax currently remains poorly studied” perhaps replace with “The phylogenetic affinities of the genus Coryphophylax remain poorly studied”?

Line 269: “The only study that assessed the phylogenetic placement based on molecular data of the genus was by Pal et al. (2018)” perhaps replace with “The only study that assessed the phylogenetic placement of the genus based on molecular data was by Pal et al. (2018)”?
I do not have S. Krishnan’s MSc thesis (and have no idea of the whereabouts of that person), but I guess that his work involved molecular data based on all the samples he deposited in GenBank. I am a bit surprised that these results are not discussed in your paper, did S. Krishnan reach the same conclusions?

Lines 271-272: “The present work, too, suffers from a similar limitation in molecular data, likewise supported by 16S rRNA data only” replace with “The present work also suffers from the limited molecular data currently available (sequences of a fragment of 16S rRNA only)”

Lines 273-281: please revise English grammar here, this part of the text is not clear. The whole Discussion is uneven in terms of language and clarity, the last author should definitely revise it.

Lines 273-274: “However, it is of merit to present a snapshot of the diversity of the lizards across these islands and confirm that “C. subcristatus” comprises multiple species”.
I found myself wondering if sp. 1 from Car Nicobar was identified as subcristatus before your paper. I can’t find that information in your text. For the sake of clarity, I think that your Table 1 should include previous identifications (vs. your new results) as well.

Line 302: “Molecular data suggest” not “Molecular data suggests”

Caption of Table 2: “Un-corrected sequence divergence”, replace with “Uncorrected p genetic divergence”

Figure 1 (and Table 1): one sample is reported from Camorta Island, but there is no collection locality for that island in Figure 8

---

## Round 0.2 · Minor Revisions

Dear Dr. Mirza, I ask you to make the necessary corrections to the manuscript so that it can be approved for publication.

·

Basic reporting

I strongly suggest all authors to double check necessary, latest and incorporated data files in MS, pdf and excel sheet. These are the following points that I feel to address and are as below:

1. Some inadvertent mistakes were observed in both MS and pdf files example; italics, conjoined words, values and units etc. were to be taken care with great priority else the essence of basic report tends to loss midway. So here, earnest request to all the author/co-authors should kindly considered from their ends (All have been highlighted/marked in color within the files).

2. Morphometric measured values considered were to the nearest 0.1 mm, but in the main text and excel sheet provided (marked in different colors) need conformity (Say if 70.8 mm then authors represent values throughout the main text and tables in uniformity.

3. Which one Comorta Island or Camorta Island ('o' or 'a')?

4. Suggesting all authors to check the arrangement for reference according to journal format.
.

Experimental design

Good.

Validity of the findings

Good.

Additional comments

I suggest authors to kindly double check every provided comments again highlighted in MS, pdf & excel sheet which itself is explanatory. The manuscript has significantly improved. However, necessary minor revisions need to be incorporated from the authors before acceptance.

Warm regard and best wishes!

Reviewer 3 ·

Basic reporting

no comment

Experimental design

no comment

Validity of the findings

no comment

Additional comments

This is my second review of the manuscript and I am satisfied with most of the changes made by the authors. Only one additional comment, in their rebuttal letter the authors wrote "The phylogeny cannot be compared as the methods were different". This is of course wrong. If we could only compare phylogenies that use the exact same methods we would be extremely limited. Also, comparing ML and Bayes topologies (for instance) is standard procedure to check for discrepancies.

---

## Round 0.3 · Major Revisions

Dear Dr. Mirza, I ask you to correct the shortcomings pointed out by reviewer 2. Without a detailed analysis of the problem in the response to the reviewer and in the manuscript, the article cannot be published.

·

Basic reporting

Good.

Experimental design

Good.

Validity of the findings

Good.

Additional comments

Minor revision needed and after corrections are checked and done from the author/co-authors end, then it can proceed to editor's final decision or acceptance of manuscript.

1. Few corrections needed are highlighted against the lines no. in yellow color and comments.

2. In the Table 1 all the specimens of C. maximiliani 22 provided from different localities are and that of C. subcristatus is 19. But for Table 2 specimens considered were more by 1 in C. maximiliani (23) and less by 2 in C. subcristatus (17) against the Table 1. It doesn't match comparison between these two tables. Please, kindly justify/very why is it so? (Comment was already provided under first review as well).

3. The rest are perfect in the manuscript.

Warm regards and my best wishes to authors for official publication soon. Thank you!

Reviewer 2 ·

Basic reporting

No Comments

Experimental design

No comments

Validity of the findings

No comments

Additional comments

The major issues I raised during the last round of review remain unanswered. I cannot therefore recommend this manuscript for publication. At the very least, I would expect a far more detailed discussion on the major pitfalls of the study when trying to reconcile the data with the results and inferences the authors make.

---

## Round 0.4 · accepted · Accept

Dear Dr. Mirza, I congratulate you on the acceptance of this article for publication. I hope that you will continue to study island populations of reptiles and send more articles to our journal.